# Physics-Informed Neural Network for Quantifying Time-Encoded Arterial Spin Labeling: A Simulation Study

**Alessandro Giupponi**[*][1] iD          ALESSANDRO.GIUPPONI@PHD.UNIPD.IT

**Chiara Da Villa**[*][1]                CHIARA.DAVILLA@STUDENTI.UNIPD.IT

**Mattia Veronese**[1,2]                MATTIA.VERONESE@UNIPD.IT

**Marco Castellaro**[1] iD              MARCO.CASTELLARO@UNIPD.IT

[1] *Department of Information Engineering, University of Padova, Padova, Italy*

[2] *Neuroimaging Department, IoPPN, King's College London, London, UK*

**Editors:** Accepted for publication at MIDL 2025

## Abstract

Arterial Spin Labeling (ASL) MRI enables non-invasive quantification of cerebral perfusion. Hadamard time-encoding improves acquisition efficiency and allows the simultaneous estimation of cerebral blood flow (CBF) and arterial transit time (ATT) via the Buxton model. Physics-informed neural networks (PINNs) integrate physical laws into neural networks, improving parameter estimation under noisy and sparse data conditions. We propose a two-stage PINN framework trained on synthetic ASL data from the Boston ASL Template and Simulator. Leveraging coupled neural networks and differential equation constraints, our method produces smoother and more robust CBF and ATT maps compared to regularized nonlinear least squares (NLLS), demonstrating its potential for clinical ASL quantification. While this work focuses on simulation data, it represents a first step toward extending such models to in vivo applications using a similar architecture.

**Keywords:** Arterial Spin Labeling, Physics-Informed Neural Network, Hadamard Encoding, Cerebral perfusion

## 1. Introduction

Arterial Spin Labeling is a non-invasive MRI technique that quantifies cerebral perfusion by magnetically labeling arterial blood water. Time-encoded pseudo-continuous ASL (te-pCASL) through Hadamard-encoded sub-boluses increases signal efficiency and enables simultaneous estimation of cerebral blood flow and arterial transit time (Woods et al., 2024). The Buxton model describes ASL signal dynamics and enables the extraction of physiological parameters such as CBF and ATT (Buxton et al., 1998). Conventional methods like nonlinear least squares are sensitive to noise and initialization, often producing unstable or noisy parameter maps. Physics-informed neural networks have emerged as a more robust alternative by embedding physical laws into deep learning, enabling both differential equation solving and parameters estimation (Karniadakis et al., 2021). PINNs improve predictive accuracy, particularly in scenarios with limited or noisy data, and increase generalization and interpretability in scientific computing tasks. Medical applications span cardiovascular modeling (Raissi et al., 2019), perfusion CT (de Vries et al., 2023), and ASL in infants (Galazis et al., 2025). Here, we extend the PINN framework proposed by de Vries et al. (2023) to te-pCASL using synthetic data from a Buxton-based simulator.

---

[*] Contributed equally

## 2. Materials and Methods

**Data Simulation**. Synthetic te-pCASL signals were generated using the Boston ASL Template and Simulator (Taso et al., 2022), which provides spatially varying ground-truth maps for CBF, ATT, and equilibrium magnetization ($M_0$). These maps were resampled to match the in-plane resolution of real ASL datasets ($3\times3\ mm^2$). To mimic real-world conditions, Gaussian noise was added following established methodologies (Bladt et al., 2020), using two plausible values of temporal signal-to-noise ratio (tSNR): 1.5 and 0.5.

**PINN Architecture**. Our model consists of two coupled Multi-Layer Perceptrons with sinusoidal activation (SIREN), which were trained on a central slice of the simulated volume:

- A **data-fitting network** $f_{tissue}(t, x, y; \phi)$ (7 layers, 256 units) receiving time and spatial coordinates (x, y) to fit the ASL signal.

- A **physics-based network** $f_{ode}(x, y; \xi)$ (4 layers, 128 units) predicting CBF and ATT from spatial coordinates.

Training minimizes a hybrid loss combining the mean square error (MSE) between observed and predicted signals and the MSE between the time derivatives of $f_{tissue}(t, x, y; \phi)$ (via automatic differentiation) and the Buxton model derivatives given the estimates of CBF and ATT. To reflect the dependence of bolus duration on acquisition time in the absence of an analytic expression, a fifth-order spline was used, and its derivative was incorporated into the Buxton model derivative to better capture time-encoding dynamics. To ensure training stability, convergence, and optimal performance, we selected loss weights of 1 and 0.5 respectively through a grid search. Optimization used Adam (learning rate: $10^{-4}$) over 30,000 epochs with batch size of 150. Only $f_{ode}(x, y; \xi)$ is retained at inference to generate CBF and ATT maps.

**Baseline.** To evaluate the proposed PINN, we compared it against a regularized NLLS approach based on the Buxton model. This method estimates CBF and ATT voxel-wise by minimizing the residuals between the observed ASL signal and the model, with a small penalty on ATT deviations from its initial guess. Optimization was performed with soft-L1 loss and physiologically plausible bounds, using the Powell's hybrid method. This serves as a conventional yet robust benchmark to assess the accuracy and spatial consistency of PINN predictions.

**Evaluation**. Performance was measured using the Structural Similarity Index (SSIM), Pearson Correlation Coefficient (PCC), and Percentage Relative Error (PRE) maps to assess spatial accuracy, correlation with ground truth, and estimation bias, respectively.

## 3. Results

PINN and NLLS achieved comparable accuracy in CBF estimation at higher tSNR (1.5), while PINN substantially outperformed NLLS in ATT estimation (Table 1). As the noise level increased (tSNR = 0.5), the performance gap widened in favor of PINN for both CBF and ATT, with notably higher SSIM and PCC values.

As shown in Figure 1, PINN produced smoother and more physiologically plausible maps. While it systematically overestimated ATT, it better preserved the clinically relevant spatial

| tSNR | | SSIM | | PCC | |
|------|-----|-------|-------|-------|-------|
| | | PINN | NLLS | PINN | NLLS |
| 1.5 | CBF | **0.825** | 0.786 | **0.969** | 0.944 |
| | ATT | **0.951** | 0.467 | **0.995** | 0.761 |
| 0.5 | CBF | **0.643** | 0.533 | **0.907** | 0.841 |
| | ATT | **0.842** | 0.212 | **0.980** | 0.482 |

Table 1: SSIM and PCC obtained in CBF and ATT estimation by the two different approaches, with tSNR of 1.5 and 0.5.

patterns and demostrated increased robustness to noise. In contrast, NLLS yielded noiser estimates with higher spatial variability, especially for ATT.

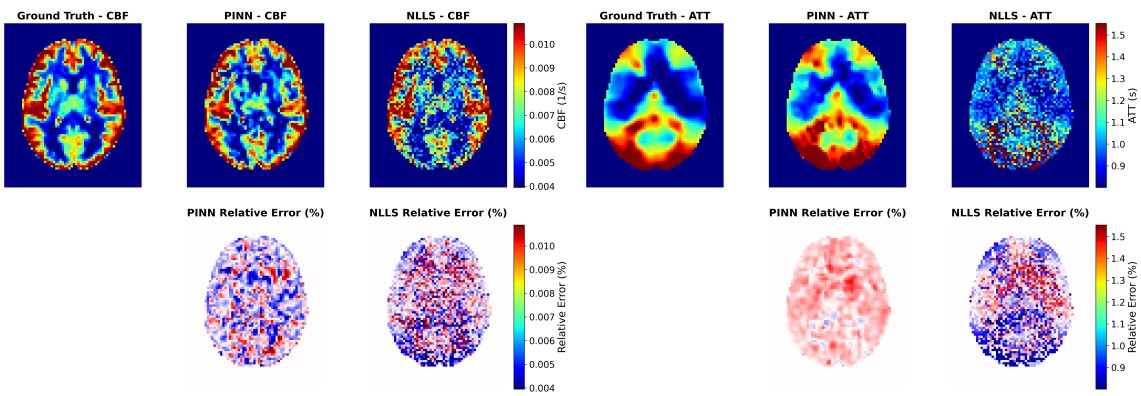

Figure 1: Comparison of CBF and ATT estimations using PINN and NLLS. The first row shows the estimated maps, while the second row presents the corresponding Percent Relative Error maps.

## 4. Discussion and Conclusion

We propose a PINN framework that effectively integrates prior knowledge from the Buxton model for ASL-based perfusion quantification. By coupling a signal-fitting network with physics-based network under hybrid loss, our model achieves robust estimation of CBF and ATT across two different noise conditions. Compared to NLLS, PINNs demonstrated superior stability in ATT estimation and comparable performance in CBF recovery, especially under low tSNR. The improved robustness stems from the model's ability to regularize solutions with physical constraints. Limitations include the use of simulated data and restriction to a single 2D slice. Future work will extend validation to in vivo data and explore 3D spatial modeling. With further refinement, PINN-based models could become valuable tools for robust, data-efficient quantification in clinical ASL imaging.

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
