# OpenReview forum: "Physics-Informed Neural Network for Quantifying Time-Encoded Arterial Spin Labeling: A Simulation Study"
_MIDL.io/2025/Short_Papers — MIDL 2025 - Short Papers_

### Official Review · Reviewer_2J7a · 2025-04-28

**Rating:** 3
**Confidence:** 4

**Summary:**

This paper proposes a PINN framework for estimating cerebral blood flow (CBF) and arterial transit time (ATT) from synthetic ASL data. While the topic is relevant, the implementation falls short of expectations for a physics-informed model: the physical constraint is enforced only through derivative matching rather than solving a full PDE, and the architecture separates data fitting and physics modeling without justification. Moreover, the absence of a hyperparameter to balance the hybrid loss raises concerns about the stability and generality of the approach. The study is limited to synthetic 2D data with no validation on real images. Overall, the contribution is too preliminary and methodologically weak to be accepted in its current form.

**Strengths:**

The topic is relevant, addressing robustness issues in ASL quantification under noisy conditions.
The use of synthetic data from a controlled simulator provides a reproducible benchmark.
Results indicate improvements over a regularized nonlinear least squares (NLLS) baseline, especially for ATT estimation under low SNR.

**Weaknesses:**

The physical constraint is only weakly enforced via derivative matching rather than solving a full PDE or ODE, limiting the "physics-informed" nature of the approach. A more rigorous physics enforcement would be expected in a PINN framework.
The separation between a data-fitting and a physics-based network complicates the method without clear benefit. No justification is provided for not using a single network with a global hybrid loss.
The hybrid loss combines data and physics terms, but no balancing hyperparameter is introduced or discussed. This is a major omission: tuning the relative weight between data and physics is standard in PINNs to ensure convergence and stability.
The study is restricted to synthetic 2D slices with no experiments on real (in vivo) data or on full 3D volumes. As a result, the clinical relevance and generalizability of the approach remain highly speculative.

---

### Decision · Program_Chairs · 2025-05-01

Accept